# Buffering the Effects of Burnout on Healthcare Professionals’ Health—The Mediating Role of Compassionate Relationships at Work in the COVID Era

**DOI:** 10.3390/ijerph19158966

**Published:** 2022-07-23

**Authors:** Ilaria Buonomo, Paolo Emilio Santoro, Paula Benevene, Ivan Borrelli, Giacomo Angelini, Caterina Fiorilli, Maria Rosaria Gualano, Umberto Moscato

**Affiliations:** 1Department of Human Sciences, LUMSA University, 00193 Rome, Italy; i.buonomo1@lumsa.it (I.B.); benevene@lumsa.it (P.B.); g.angelini@lumsa.it (G.A.); fiorilli@lumsa.it (C.F.); 2Department of Life Sciences and Public Health, Catholic University of the Sacred Heart, 00168 Rome, Italy; paoloemilio.santoro@unicatt.it (P.E.S.); umberto.moscato@unicatt.it (U.M.); 3Department of Women, Children and Public Health Sciences, Fondazione Policlinico Universitario Agostino Gemelli IRCCS, 00168 Rome, Italy; 4Department of Public Health Sciences, University of Turin, 10124 Turin, Italy; mariarosaria.gualano@unito.it

**Keywords:** healthcare workers, compassion at work, burnout, COR theory, occupational medicine

## Abstract

Managing the COVID-19 pandemic posed several challenges for healthcare professionals, which likely heightened their risk of burnout (Amanullah and Ramesh Shankar, 2020) and, consequently, their general physical and mental health. Although it may not be possible to address and eliminate the causes of burnout, current research informs healthcare organizations about protective strategies to reduce its detrimental consequences. The promotion of compassionate interactions among healthcare professionals may play such a role. Compassion within healthcare organizations positively affects individual performance and well-being. Building on these considerations and within the framework of the Conservation of Resources theory, this study explores the relationships among burnout dimensions, received compassion at work, and general health in 711 Italian healthcare professionals (68.5% female), aged between 21 and 73 years (M_age_ = 36.4, SD = 11.2). Analyses were conducted to investigate the association between burnout and general well-being (H1) and between burnout symptoms and perceived compassion at work (H2); and the mediational role of compassion in the relationship between burnout symptoms and general well-being. H1 and H2 were confirmed (*r* < 0.01 for both), and a SEM model showed the mediating role of compassion at work in the association between burnout symptoms and general well-being (RMSEA < 0.08, SRMR < 0.08, CFI and TLI > 0.90). Theoretical and practical implications of the findings are discussed in the paper.

## 1. Introduction

This work aims to explore the role of compassion in promoting well-being in healthcare professionals during the COVID-19 pandemic. Identifying protective factors that could support healthcare workers during and after these types of events are crucial for their job performance and mental well-being. This category of professionals, indeed, was challenged by the management of the COVID-19 pandemic, with detrimental effects on burnout and general health conditions [1]. Interestingly, the first reviews on this topic did not find a clear link between the exceptional job demands posed by the pandemic and burnout conditions [1].However, more recent studies reported a recurring association between working as a healthcare professional during the pandemic and more intense feelings of fear, frustration, and anxiety, even linked to the risk of being infected and spreading the virus to loved ones [2].

While the link between job challenges and heightened stress and fear reenacts what was already observed during the SARS and swine flu pandemics [3,4], the worldwide impact of the COVID-19 pandemic had a more intense and widespread effect. Such pervasive conditions made it more difficult for healthcare organizations to intervene directly in the causes of burnout: longer shifts, shorter breaks, and higher emotional stress, as the conditions in which the workers operated were not negotiable during the acute phases of the pandemic [1,5,6]. At the same time, these experiences still impact the work and well-being of healthcare employees [7,8,9]. Thus, it is important to individuate effective protective factors that could promote well-being and buffer the detrimental effects of emergency-related job conditions and requirements. This process starts from the acknowledgment of the effects of the pandemic on this specific group of professionals.

Research is starting to address the long-term consequences of COVID-19. For example, changes have been reported in individual perceptions of work and organizations and the link between work and personal life due to the pandemic-related experiences at work [10,11]. The focus is on how employees perceive their job conditions and also the global pandemic-related organizations. During the COVID-19 outbreak, indeed, employee perceptions have been crucial for organizational management. It has been argued that employee perceptions of the pandemic are a critical point for managers to shape followers’ attitudes and behaviors at work and, consequently, maintain stable productivity and effectiveness [2].

At the same time, studies on employee well-being during the pandemic [12,13,14,15] and even before the COVID-19 outbreak [16,17,18,19], conclude that employee perceptions of their organizational environments have a crucial role in preventing burnout and promoting well-being. According to these studies, elements such as trust [20,21], social support [22,23], good communication with colleagues and management [24,25], and effective teamwork [26,27] have a significant impact on burnout risks and personal well-being. In addition, such associations have been reported in samples of healthcare workers [28,29,30,31]. By acknowledging the importance of such protective factors, current research may provide compelling indications to healthcare management to reduce the potentially detrimental consequences of long-term COVID-19. It urges, indeed, to implement programs aiming to take care of healthcare professionals, not only to maintain their productivity but also their physical and mental health.

In this respect, recent psychological literature identified compassion at work as a crucial protective factor for employee well-being in the workplace and in their whole life [32,33,34,35,36,37,38,39,40].

Compassion is described as a “sensitivity to suffering in self and others with a commitment to try to alleviate and prevent it” [41]. As reported by Dutton and colleagues, suffering is a common condition at work that could emerge as negative emotions, stress conditions, or burnout [32,33,42]. While the reasons for such suffering may be due to employee personal life events [32] or poor relationships at work [43], several studies have shown that significant organizational changes [44] and job roles implying caregiving tasks [45] might heighten the likelihood of experiencing some suffering at work. According to these studies, healthcare professionals may be specifically entitled to benefit from compassionate work environments.

Building on international data on the COVID-19 experience in healthcare organizations and on research findings on potential risks and protective factors, this study aims to verify the associations among burnout, compassion at work, and well-being in a sample of Italian healthcare workers one year after the onset of the COVID-19 pandemic. More specifically, as detailed below, the study aims to verify whether and how the perception of working in a compassionate environment may benefit healthcare workers by buffering the effects of burnout symptoms on general well-being.

## 2. Theoretical Background

The effects of detrimental and protective factors on healthcare professionals’ well- being may be explained in the light of the Conservation of Resources (COR) theory [46,47]. According to the COR theory, social and environmental conditions may foster or counteract personal well-being, depending on the degree of resources they allow the individuals to gather and transfer from one domain of life (e.g., work) to the other (e.g., personal life) [48]. Resources include, for example, objects, personal characteristics, money, knowledge, time, and external conditions. Stress and burnout conditions occur from the loss of such resources, while well-being corresponds to their gathering and meaningful use in any domain of life. The COVID-19 outbreak and its consequences constituted a prolonged loss of resources and posed significant challenges to resource gathering processes. While this is true for any person experiencing the outbreak [49,50,51,52], people who worked on the frontline in managing the pandemic may have suffered directly from the loss (or the potential loss) of these types of resources. Among such effects, burnout emerges as a potential form of resource loss for healthcare professionals [6,8,53].

### 2.1. Stress and Burnout Risks in Healthcare Professionals

Burnout is a syndrome experienced by many professionals engaged in highly demanding jobs; among these, healthcare professionals have an elevated risk of suffering this syndrome. The syndrome is due to prolonged exposure to stressful events [54]. Longitudinal studies have shown that the emotional suffering of a burned-out worker is the prelude to other kinds of symptoms [55]. Associated with high emotional exhaustion are symptoms like cynicism toward work, detachment from work-related relationships, a sense of inefficacy, and negative coping strategies [56]. Recently, Schaufeli and colleagues [57] have built a multidimensional tool to evaluate further latent factors, including emotional and cognitive impairments, which were confirmed in Italian samples [58,59].

Concerning burnout among healthcare workers, West et al. [54] claimed that physicians’ emotional exhaustion includes “feeling used up at the end of a workday and having nothing left to offer patients from an emotional standpoint” (p. 516). Importantly, low performance, generally associated with burned-out workers, is mainly studied in medical professions (for a review, see [60]). The research, focused on physician-reported errors and their suboptimal patient care, has shown strict relationships between surgeons’ work-related stress and medical errors [61]. Nowadays, physicians’ burnout is a public health issue broadly impacting society. Specifically, it has negative impacts at three levels: the individual and familial level, the patient treatment level, and at the level of costs to the healthcare system.

The job demands-resources model [62] represents an approach to understanding and evaluating workers’ burnout risk, including that of healthcare professionals [53]. The same stressful events (demands) may find a different solution from the worker (resources). According to the model, work stress may occur when there is an imbalance; thus, job demands overcome people’s resources. Focusing on the latter, both internal and external resources can help workers cope with stressful events [63]. Recently, several scholars have analyzed how enhancing resources may prevent and mitigate burnout among physicians (for a review, see [55]). Among others, adequate internal resources include emotional skills, communication strategies, self-efficacy, mindfulness, self-compassion, and subjective well-being [54]. Similarly, enhancing workers’ external resources may ameliorate the work environment for all workers [64]. In this regard, factors such as cooperation among colleagues, positive leadership, organizational support systems, and organizational well-being systems are particularly effective in preventing burnout conditions.

Overall, due to the well-demonstrated implications of healthcare workers’ burnout, it is crucial to investigate which factors promote workers’ awareness and support their aid requests as soon as they manifest the first symptoms [55]. It is a cogent issue considering the risk of poor and inefficient medical services negatively impacting the care of patients.

### 2.2. The Detrimental Effects of Burnout on General Well-Being

Burnout conditions impact employees’ work life and general well-being. Several studies have shown that burnt-out and exhausted individuals report lower general health levels from both physical [65,66] and psychological [67] perspectives. This link has been confirmed in healthcare professionals [68,69,70,71,72].

Hobfoll [46,47,73] has broadly explained the reasons behind the impact of the work environment on personal life. According to the COR theory, indeed, personal life, work-life, social, and cultural participation are perceived as separate but strongly intertwined dimensions of one’s life. Indeed, all these contexts are intertwined so that people can gain, lose, and use any resource (e.g., time, money, knowledge, relationships, personal characteristics) in any context. In other words, according to the COR theory, the resources built or lost in the work environment will influence not only the work experience but also the life of employees in general. According to the COR theory, this mechanism is possible because losses spirals: when people lose resources, they are also more exposed to future loss. This mechanism perpetuates itself as long as the individual cannot obtain resources that would allow a resource gain. For example, meeting a supportive colleague during a frustrating day at work may help a healthcare worker feel relieved and uplifted or find a solution for the problem he or she was dealing with. This interaction, in turn, will lower the frustration, benefit the worker’s coping strategies, and, finally, impact his or her general well-being. Building on these considerations, it is crucial to identify valuable resources at work (i.e., protective factors) that may foster resource gain spirals in healthcare workers. Considering the COR-based approach and the COVID-19 outbreak [51], this is especially crucial for healthcare workers dealing with the management and consequences of the pandemic. According to the most recent integration of the COR theory, namely the crossover model [73,74], psychological states are transmitted across individuals so that relationships at work allow for resource gains. Therefore, compassion at work may act as a valuable mechanism through which employees interact to exchange and build significant resources for their work and well-being.

### 2.3. The Protective Role of Compassion at Work

As stated above, compassion emerges when someone in the environment expresses suffering (e.g., negative emotions, stress, burnout, frustration) [42,75]. Compassionate behaviors are expressed through four dimensions [76]: (1) attending to the other’s suffering; (2) understanding what is causing the other’s distress, possibly by confronting and sharing experiences with him or her; (3) empathizing with the other’s suffering and helping her or him to elaborate on the feelings, the thoughts and the contextual conditions that contributed to the onset of such suffering; and (4) helping others by appropriate, skilled, and thoughtful actions with the intent of reducing the suffering or, at least, providing others with coping strategies. Differently from empathy, compassion implies a motivational dimension related to the urge to help others, or at least do something to their advantage in a suffering condition other than merely tuning in with their emotions [77]. Furthermore, the focus on the ability to choose helpful actions that are valuable and pertinent in the context allows compassionate employees to support their colleagues or followers while keeping in mind work tasks and objectives [37].

Promoting such behaviors at work has several advantages for the quality of work and well-being. For example, West and colleagues showed that healthcare professionals are more effective when leaders are compassionate, whereas they perform worse when they perceive high degrees of control and coercion (West et al., 2014). Furthermore, when healthcare workers feel their work and needs are acknowledged and valued, their satisfaction, commitment, and sense of belonging increase [37], they make fewer mistakes and report fewer relational problems with colleagues [78,79]. At the same time, experiencing compassion at work has positive effects on personal well-being. It positively impacts positive affects and emotions [32,80,81], psychological well-being and mental health [82,83], and perceived general health [84].

Overall, current research informs that compassion at work could be a valuable mediating variable between burnout symptoms and general well-being. From one side, indeed, it is activated by acknowledging other people’s suffering (expression of burnout symptoms). Thus, people expressing burnout signs are more likely to encounter a compassionate response from their work environment. On the other side, receiving compassion benefits general well-being conditions.

### 2.4. The Present Study

Building on the theoretical frameworks and the research findings reported above, this study aims to verify whether and how perceived compassion at work mediates the link between burnout symptoms and general well-being in a sample of Italian healthcare workers dealing with the management and the consequences of the pandemic.

More specifically, we propose three hypotheses (See Figure 1).

**Hypothesis** **1.**
*There is an association between burnout symptoms and general well-being.*


**Hypothesis** **2.**
*There is an association between burnout symptoms and perceived compassion at work.*


**Hypothesis** **3.**
*There is a mediating effect of compassion at work in the relationship between burnout symptoms and general well-being.*


Furthermore, data on how the variables of interest vary between healthcare workers who contracted and did not contract COVID-19 since the beginning of the pandemic will be examined.

## 3. Materials and Methods

### 3.1. Participants

The participants in the study were 711 Italian healthcare workers (68.5% female), aged between 21 and 73 years (M_age_ = 36.4, SD = 11.2). The inclusion criteria for the study were: (1) being an Italian healthcare worker; and (2) voluntarily agreeing to participate. Forty-seven percent of participants were physicians, and 35.8% were nurses. The remaining 17.2% included psychologists and psychotherapists, pharmacists, students, and other healthcare roles. Finally, 14.4% of participants had contracted COVID-19 since the beginning of the pandemic, and 91.8% were vaccinated against COVID-19.

Although previous studies, e.g., [84,85], addressed the association between stressful job conditions and experienced compassion in healthcare workers, to the best of our knowledge, the involvement of this sample is particularly valuable for three reasons: (1) differently from previous studies, it considers the role of compassion received from colleagues and leaders in this category of workers; (2) it implements a new tool to measure burnout in healthcare professionals (i.e., the Burnout Assessment Tool described in the Section 3.3), thus allowing practitioners to gather data about its use in healthcare contexts; and (3) it allows for a comparison between healthcare workers and other professionals involved in similar studies with the same burnout tool e.g., [57].

### 3.2. Procedure

This study was carried out in Italy between May and June 2021 using a cross-sectional design and a convenience sample among workers of hospital institutions. The data was collected online via the G Suite Google Platform, which did not allow the respondent to proceed if the fields were not completed. For this reason, there was no missing data. The link to the survey was sent via a hospital mailing list addressed to HealthCare Workers (HCWs) interested in participating in the study. The participants were informed of the research objectives, that the answers would remain anonymous, and that no personal data was acquired. This study was conducted under the requirements of privacy and informed consent laid down by current Italian law (Law Decree DL-196/2003), approved by the Ethics Committee for Scientific Research (CERS) of LUMSA University conducted under the Declaration of Helsinki of 1964, and its latest versions.

### 3.3. Measures

Burnout Assessment Tool (BAT [56,57,59]): the BAT comprises two dimensions and six subscales. The BAT-C dimension includes 23 items, divided into four core subscales: exhaustion (EXH, 8 items), mental distance (MD, 5 items), cognitive impairment (CI, 5 items), and emotional impairment (EI, 5 items). The BAT-S includes 10 items measuring secondary symptoms: psychological distress (5 items) and psychosomatic complaints (5 items). All 33 items were scored on a five-point Likert scale, ranging from “never” (1) to “always” (5). The answers were summed and averaged resulting in a score range from 1 to 5. The total BAT scores can assess the level of burnout, and the independent scores on its six dimensions (core symptoms and secondary symptoms) can provide more information. In this study, α = 943. For the scope of our study, only the BAT-C dimension was considered in the model. For this dimension, α = 940.

The World Health Organization Well-Being Index [86,87,88] includes five items that evaluate the global assessment of well-being (Topp et al., 2015). The scale is rated on a six-point scale, from “all of the time” (0) to “at no time” (5), with scores ranging from 0 (absence of well-being) to 25 (maximal well-being). Higher scores indicate an increased sense of well-being. In this study, α = 879.

Compassion at Work [32] was measured with a three-item scale rated on a five-point Likert scale (1 “never” to 5 “very often”). The scale investigates experienced compassion (a) on the job, (b) from one’s supervisor, and (c) from one’s coworkers. In this study, α = 831.

### 3.4. Plan of Analysis

Regarding the data, three procedures of data exploration were applied: (a) uni- and multivariate outlier analysis (the Mahalanobis’s distance was set to *p* < 0.001) [89]; (b) score distribution analysis (skewness and kurtosis cut-off points were set to −2 and +2, respectively) [90]; and (c) missing value analysis (missing values were skipped listwise) [91]. At the end of these procedures, we obtained the sample described above. Secondly, to test the common method variance bias [92], the Harman’s single-factor test was performed. The findings showed that the single factor emerging from the exploratory factor analysis only accounted for the 37% of the covariance among the measures, thus no issues are associated with common method variance in the data [93].

Regarding the model, two analyses were performed: a Confirmatory Factor Analysis (CFA) and a Structural Equation Modeling (SEM) [94], both with MPlus version 8 [95]. The CFA was performed to examine the measurement model. To enhance the reliability and parsimony of our model, item parcels were created for Burnout and General well-being. For Burnout, the parcels were based on the subscales, so it included four parcels (Exhaustion, Mental Distance, Cognitive Impairment, and Emotional Impairment). For General well-being, two parcels were created according to the degree of correlation of each item with the total score [96]. Parceling reduces the number of free parameters requiring estimation and reduces the sampling error sources [97,98,99]. In addition, the Robust Maximum Likelihood Approach (MLR) was used to deal with non-normality in data [100].

Next, the structural model was tested using the structural equation modeling (SEM) approach [94]. Under the model, Burnout was directly and indirectly (via Compassion at work) correlated with General well-being (see Figure 1). A multi-faceted approach to the assessment of the fit of the model [101] was used, based on the following parameters: the Chi-square likelihood ratio statistic, the Tucker and Lewis Index (TLI), the Comparative Fit Index (CFI), the Root Mean Square Error of Approximation (RMSEA) with its confidence intervals, and the Standardized Root Mean Square Residual (SRMR). We accepted TLI and CFI values greater than 0.90 [102], RMSEA values lower than 0.08 [103,104], and SRMR values lower than 0.08 [104,105]. Furthermore, we recognize that, according to Kline [102], the Chi-square index could result significance because of its sample size-sensitive bias.

Finally, because the sample is not equally distributed between healthcare workers who caught COVID-19 since the beginning of the pandemic (*n* = 102), and workers who did not (*n* = 609), and because a SEM model requires at least 150 subjects to be calculated with an acceptable degree of reliability [94,106], it was not possible to compare and contrast the SEM model described above within the two sub-samples. Nevertheless, data on burnout, compassion at work, and general well-being have been explored in the two sub-samples to provide a higher degree of detail on the role of COVID-19 infection. More specifically, ANOVA tests were calculated to verify significant differences in the means of burnout, compassion at work, and general well-being in the two groups, and correlations were calculated, other than in the whole selected sample, in the two subgroups as well.

## 4. Results

### 4.1. Measurement Model

The measurement model showed a good fit to the data: χ^2^(24) = 103.033, *p* = 0.000, CFI = 0.970, TLI = 0.955, RMSEA = 0.068 (90% CI = 0.055–0.081, *p* = 0.013), SRMR = 0.042. Table 1 shows the AVE and Composite reliability coefficients are above 0.65 and 0.85, respectively. Table 2 shows the correlations among Burnout, Compassion at work, and General well-being. As expected, Burnout was inversely correlated with the other variables, while Compassion at work and General well-being were positively associated. The means of BAT-C, General well-being, and Compassion at work did not change significantly between workers who contracted vs. did not contract COVID-19 since the beginning of the pandemic (*p* > 0.05 in the three ANOVA tests conducted). Interestingly, in the sub-sample who contracted COVID-19, burnout and compassion at work were not significantly related (r = −0.130, *p* > 0.05).

### 4.2. Final Model

The final model (shown in Figure 2) showed a good fit to the data: χ^2^(24) = 103.033, *p* = 0.000, CFI = 0.970, TLI = 0.955, RMSEA = 0.068 (90% CI = 0.055–0.081, *p* = 0.013), SRMR = 0.042.

The model showed that Burnout was negatively associated with General well-being (b = −0.531, *p* < 0.001) and with Compassion at work (b = −0.320, *p* < 0.001), thus confirming H1 and H2. Furthermore, the model confirmed the mediational role of Compassion at work in the link between Burnout and General well-being (bDIRECT = −0.531, *p* = 0.000, bINDIRECT = −0.089, *p* = 0.000), showing a partial mediation and, thus, confirming H3.

The percentages of variance explained were 10.3% for Compassion at work and 45.4% for General well-being.

## 5. Discussion

The findings show that receiving compassion at work partially mediated the association between burnout symptoms and general well-being in a sample of Italian healthcare workers, thus confirming all the formulated hypotheses.

Considering H1 (We expected to find an association between burnout symptoms and general well-being), we confirmed previous literature on the role of job-related burnout symptoms in perceived general well-being. This result is consistent with the principles of the COR theory [46,47,73], according to which, resource gains and losses spread across different domains of life.

Considering H2 (We expected to find an association between burnout symptoms and perceived compassion at work), we confirmed previous literature about the role of suffering in activating compassionate behaviors [42], as well as the possibility of reframing burnout conditions as a form of suffering at work [33,42,107,108]. Furthermore, the data in this study were gathered during the COVID-19 pandemic (more specifically, after the second wave). Thus, they were likely influenced by the long-term strains occurring from the management of the pandemic in the work and personal life of the participants. According to the event system theory, the more individuals perceive an event as “strong”, the more they are motivated to behave in a certain way [109]. Event strength can be defined according to its novelty (how much the event was foreseeable before its onset), disruption (how much it impacts the external environment), and criticality (how much it is important or crucial) [109]. According to these criteria, the pandemic is categorizable as a new, disruptive and critical event [110,111]. The high degree of saliency of the pandemic event likely triggered the onset of compassionate behaviors at work, consequently influencing our participants’ perceptions of receiving compassion. A recent longitudinal study [112] used this framework to show that the ability of teams to provide compassionate care for their members mitigated the effects of the perceived strength of the pandemic event on individual suffering. Although this study did not include healthcare workers, it is a valuable contribution to the debate around the detrimental impact of the pandemic in organizational contexts and the triggering effect for a compassionate response from colleagues.

Finally, in relation to H3 (We expected to find a mediating effect of compassion at work in the relationship between burnout symptoms and general well-being), we showed that experiencing compassion at work may partially buffer the effects of burnout symptoms on general well-being. This is further confirmation of the crossover model [73,74], which states that individuals interact to exchange and build significant resources. Furthermore, our findings are consistent with the PERMA model of well-being [113], according to which positive emotions (E) and positive relationships (R) are two main aspects of well-being conditions. Consistently receiving compassion was shown to have positive effects in terms of the feelings of safety and protection perceived at work and the emotional balance of people receiving compassionate care [32,33,78]. As reported by Lilius and Dutton, receiving compassion at work improves well-being and provides meaning to suffering (stress, frustration, burnout) in personal life and in the context of job demands [32,42]. It seems likely that the ability of compassion to create shared meanings around adverse events or mental states at work is a crucial part of the explanation for the mediating role of compassion in the link between burnout and well-being. Meaning is another dimension of well-being in the PERMA model [113] and its application to the organizational field [114,115]. The sensemaking process occurring during compassionate interactions requires workers to acknowledge the role of other organizational actors (other colleagues, leaders, managers) and of the organization as a system in at least two processes: the perceived contribution to the suffering and the ability to sustain compassionate processes. Considering the first process, the Job Demands-Resources model [62] explains how organizations might be perceived as a source of stress and suffering by employees, for example, by setting high challenges for employees but not providing the needed resources to respond to these challenges. Although the pandemic has been an exceptional, unforeseeable event, organizational changes, whether imposed by external events or programmed by the management, are usually a source of stress and suffering for employees [12,116].

Regarding the ability of the organization to sustain compassionate processes, Atkins and Parker [74] showed that interindividual compassion in organizations does not occur in a void. On the contrary, it is embedded in several different contexts: the personal contexts of the actors directly involved in the interaction (e.g., individual differences, organizational roles), their relational contexts (e.g., their degree of perceived similarity or closeness, the social power occurring between them), and the general organizational context (e.g., the degree to which the organization supports compassionate care in the workforce, through norms, structures, practices, and leadership styles). Research on compassion in organizations is starting to address some of these dimensions, for example, by exploring the role of leadership styles on compassionate attitudes and behaviors [35,117,118], or by designing scales and questionnaires aimed at measuring different dimensions of compassion at work [32,119]. Furthermore, current research shows that when organizations acknowledge compassion as a value to be expressed through processes and practices, it benefits employee well-being, organizational performance and effectiveness, and users’ satisfaction [107,108,120,121].

## 6. Conclusions

Overall, this research extends the usefulness of the COR theory, as well as the Compassion at work framework, by revealing the potential use of these approaches in the preventive occupational assessment for professional hazards. More specifically, both could be used to evaluate jobs in relation to work-related stress risk. In addition, the association of COR-related variables and Compassion at work can be linked to health outcomes to monitor the well-being of healthcare workers.

Due to the critical role of healthcare workers’ well-being dimensions in the quality of hospital services, future research should further address this topic by analyzing training procedures to improve the management skills related to these outcomes and to prevent occupational stress.

### 6.1. Practical Implications

This study contributes to current literature reporting the protective role of compassionate behaviors in healthcare organizations, thus contributing to academic and professional training in this field. Indeed, the training of students in healthcare fields (i.e., medicine, psychology, nursing) is an important goal. Considering the large amount of research showing the protective role of compassion (and self-compassion, e.g., [122]) for healthcare professionals, it would be valuable to provide academic training on compassionate attitudes and behaviors towards patients, colleagues, and oneself as a professional. Likewise, the same type of training has shown its effectiveness among already established healthcare professionals. Although some examples of compassionate training in the healthcare field have been described in the literature [37,78,85], a proper adaptation of such contents (and related procedures) to national healthcare contexts is needed. It is likely that national and regional norms, the autonomy of each organization, specific aims, users, and the location in which the organization operates make it necessary to adapt current training procedures to fit better the specific skills needed in each context and effectively put compassion into use in workplaces.

Secondly, the protective role of compassionate behaviors discussed above as well as the role of potential COR-related issues could be harnessed to improve the tools used to evaluate work-related stress. In common tools, these aspects are poorly or not represented at all. For this reason, the practical implication of improving actual evaluation tools in the occupational field of work-related stress evaluation could enhance workers’ protection from occupational hazards, leading to improvements in their health status and well-being at work. Furthermore, developing new indicators in occupational risks assessment could lead to a new management model for total health and safety at work.

### 6.2. Limitations

Firstly, the large imbalance between participants who contracted COVID-19 and those who did made it impossible to test the associations in these two groups. Indeed, it is likely that the lack of a correlation between burnout and compassion in the subsample of workers who caught COVID-19 is due to the relatively low number of subjects in the group compared with the other. Further studies may test whether compassionate relationships have a differential impact on healthcare workers who experienced COVID-19 as patients. Considering the sample composition in terms of professional roles, the sample is mainly composed of physicians and nurses. Although this composition may mirror the composition of healthcare staff in healthcare organizations, it would be interesting to test the proposed associations in samples of different healthcare workers or compare professionals with trainees and students attending the organizations for training purposes.

Secondly, despite several previous studies acknowledging the detrimental, disruptive impact of the COVID-19 pandemic on healthcare workers, a measure of COVID-19 event strength may have clarified the role of the pandemic perception in our results.

Thirdly, the use of a non-probability sample could limit the external validity of our results.

Finally, a longitudinal study would enable further clarification of links among the studied variables.

## Figures and Tables

**Figure 1 ijerph-19-08966-f001:**
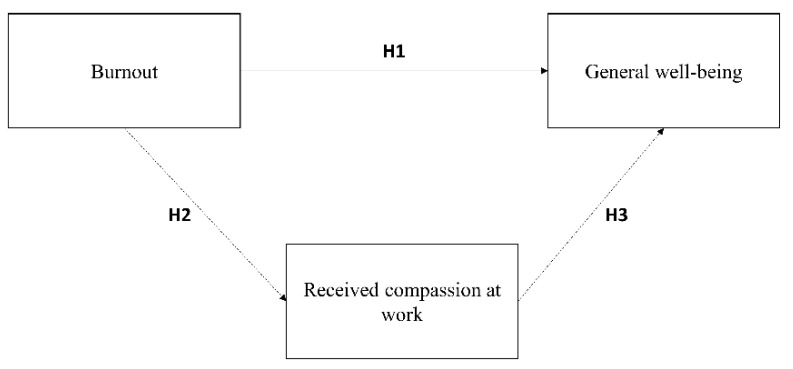
Theoretical model. Note. H3 refers to the mediating effect of compassion in the link between burnout and general well-being.

**Figure 2 ijerph-19-08966-f002:**
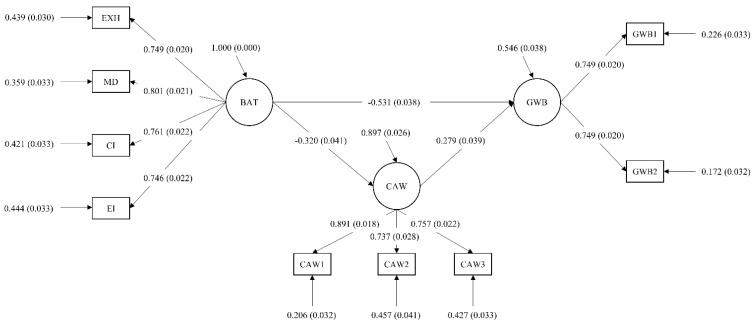
Final model. Results of the structural equation model. Standardized effects and significance values are reported. BAT = BAT-C subscales (as detailed in the parcels); CAW = Compassion at Work; GWB = General Well-Being.

**Table 1 ijerph-19-08966-t001:** Average Variance Extracted and Composite Reliability.

	AVE	CR
BAT-C	0.69	0.89
General WB	0.68	0.91
Compassion at work	0.75	0.90

Note. AVE = Average Variance Extracted, CR = Composite Reliability.

**Table 2 ijerph-19-08966-t002:** Associations among Burnout, Compassion at work, and General well-being.

	BAT-C	General WB	Compassion at Work
BAT-C	-		
General WB	−0.543 **	-	
Compassion at work	−0.268 **	0.394 **	-

Note. ** = *p* < 0.01.

## Data Availability

This published article includes all the data generated or analyzed during the study. The raw data of this research can be obtained by contacting the authors.

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
