# Peer review of "Buffering the Effects of Burnout on Healthcare Professionals’ Health—The Mediating Role of Compassionate Relationships at Work in the COVID Era"

_ijerph, 2022, doi:10.3390/ijerph19158966_

Round 1

Reviewer 1 Report

The article is interesting, and the researched problem has scientific potential. However, some problems need to be solved: 

1. The introduction should briefly place the study in a broad context and highlight why it is important. It should define the purpose of the work and its significance. Finally, briefly mention the main aim of the work and highlight the principal conclusions. I suggest splitting the first section into two sections: introduction and literature review (or theoretical background). 

2. Include a figure of the hypothesized model in section 2. 

3. The use of self-administered questionnaires can generate a problem that may affect the relevance of the research: the bias effect or common method bias - CMB (see: Podsakoff PM, MacKenzie SB, Lee, JY, Podsakoff NP. Common method biases in behavioral research: A critical review of the literature and recommended remedies. Journal of Applied Psychology. 2003; 88(5):879-903.). Such problems arise when data on independent and dependent variables emanate from the same respondent and the same measurement scale exists throughout the questionnaire. Authors must take action to prevent common method bias - CMB (e.g., https://doi.org/10.3390/ijerph182312387) 

4. Please provide measures for the validity and reliability of the SEM model (Cronbach alpha, composite reliability, AVE, etc.) 

5. In my opinion, a section of conclusions that includes theoretical and managerial implications, along with research limitations and future research directions, would be helpful. 

The paper has scientific value and can be published after carefully reviewing the reported issues. 

Author Response

Dear Reviewer 1,

Thank you for your comments and suggestions on our paper. We read them carefully and addressed to your concerns.

Below you can find your original comments and the relative point-by-point answers.

  1. The introduction should briefly place the study in a broad context and highlight why it is important. It should define the purpose of the work and its significance. Finally, briefly mention the main aim of the work and highlight the principal conclusions. I suggest splitting the first section into two sections: introduction and literature review (or theoretical background). 

We reorganized the contents in two distinct sections, as suggested. Furthermore, we opened the Introduction section with a clear statement on the main scope of the manuscript and recalled the implications of this state throughout the paper, to clarify how our work could be useful for healthcare organizations. Finally, we moved the Cor theory mentions and related paragraph in the Theoretical background section.

  1. Include a figure of the hypothesized model in section 2. 

We included the Figure.

  1. The use of self-administered questionnaires can generate a problem that may affect the relevance of the research: the bias effect or common method bias - CMB (see: Podsakoff PM, MacKenzie SB, Lee, JY, Podsakoff NP. Common method biases in behavioral research: A critical review of the literature and recommended remedies. Journal of Applied Psychology. 2003; 88(5):879-903.). Such problems arise when data on independent and dependent variables emanate from the same respondent and the same measurement scale exists throughout the questionnaire. Authors must take action to prevent common method bias - CMB (e.g., https://doi.org/10.3390/ijerph182312387

Thank you for your suggestion, we conducted the Hartman single factor test and inserted it in the Plan of Analysis section.

  1. Please provide measures for the validity and reliability of the SEM model (Cronbach alpha, composite reliability, AVE, etc.) 

Thank you for your suggestion. The original version of the manuscript included the Cronbach’s Alphas. We inserted a Table with AVE and CR data in the revised manuscript.

  1. In my opinion, a section of conclusions that includes theoretical and managerial implications, along with research limitations and future research directions, would be helpful. 

Thank you for the suggestion. We moved the implication and limitation sections in the Conclusion

Reviewer 2 Report

A good paper. Some minor ammends suggested

Line 20 - Use full text for COB

LIne 39 - suggest removal of sentence commencing line 39/ending 42. It is not a rewording of preceeding text, as suggested

Line 47 - Replace "did not allow" with "made it more difficult"

Line 80 - Unclear if author is referring to consequences of long covid 19 or long-term consequences of covid-19. I suspect it's the latter, and if so, the text requires editing

LIne 131-133. Incomplete sentence - requires rewording

LIne 416 =: Should plural of "tool" be used? 

Author Response

Dear Reviewer 2,

Thank you for your comments and suggestions on our paper. We read them carefully and addressed tyour concerns.

Below you can find your original comments and the relative point-by-point answers.

Line 20 - Use full text for COB - We modified accordingly

LIne 39 - suggest removal of sentence commencing line 39/ending 42. It is not a rewording of preceeding text, as suggested – We removed the text

Line 47 - Replace "did not allow" with "made it more difficult" - We modified accordingly

Line 80 - Unclear if author is referring to consequences of long covid 19 or long-term consequences of covid-19. I suspect it's the latter, and if so, the text requires editing - - We modified accordingly

LIne 131-133. Incomplete sentence - requires rewording – We reworded to clarify the message

LIne 416 =: Should plural of "tool" be used? 

Thank you for the suggestion. We moved the implication and limitation sections in the Conclusion

Round 2

Reviewer 1 Report

The paper ca be published in current version.